# Association between Dietary Behaviors and BMI Stratified by Sex and the *ALDH2* rs671 Polymorphism in Japanese Adults

**DOI:** 10.3390/nu14235116

**Published:** 2022-12-01

**Authors:** Maki Igarashi, Shun Nogawa, Tsuyoshi Hachiya, Kyohei Furukawa, Shoko Takahashi, Huijuan Jia, Kenji Saito, Hisanori Kato

**Affiliations:** 1Laboratory of Health Nutrition, Department of Applied Biological Chemistry, Graduate School of Agricultural and Life Sciences, The University of Tokyo, 1-1-1 Yayoi, Bunkyo-ku, Tokyo 113-8657, Japan; 2Research and Development Department, Genequest Inc., 5-29-11 Siba, Minato-ku, Tokyo 108-0014, Japan; 3Department of Genomic Data Analysis Service, Genome Analytics Japan Inc., 15-1-3205 Toyoshima-cho, Shinjuku-ku, Tokyo 162-0067, Japan

**Keywords:** precision nutrition, rs671 polymorphism, dietary behaviors, BMI, Japanese

## Abstract

The rs671 polymorphism, unique to East Asians, is well known to change the sensitivity to alcohol. Moreover, this polymorphism is associated not only with alcohol intake but also with several dietary behaviors (DBs), chronic diseases, and BMI, but the triadic association among the rs671 genotype, DBs, and BMI is unclear. This study included 12,271 Japanese subjects and aimed to observe this three-way association using the rs671 polymorphism, data of 56 DBs, and BMI. All analyses were stratified by participant sex. First, linear regression analyses resulted in significant associations between 18 and 21 DBs and BMI in males and females, respectively. Next, genetic heterogeneity was observed in all sub-groups via interaction analysis of the rs671 genotype stratified by drinking habits. Finally, we observed the characteristics of BMI-related DBs based on the rs671 genotype via stepwise regression analyses stratified by the rs671 genotype and drinking habits. Notably, positive associations were observed between *lactobacillus* beverage intake and BMI among participants with the rs671 polymorphism AA genotype in both sexes. This study suggests that the rs671 polymorphism modifies the association between DBs and BMI independently of drinking habits, providing evidence for the potential use of rs671 polymorphism information for precision nutrition with East Asians.

## 1. Introduction

“Precision nutrition”, known as “personalized nutrition”, is needed, instead of a one-size-fits-all diet, to prevent public health and disease. The research field of precision nutrition aims to understand the complex interplay among many factors, such as genetics, metabolism, the gut microbiota, food environments, and their impact on health [1,2]. This field has received worldwide attention in recent years. The National Institutes of Health (NIH) in the United States has even announced a 2020–2030 strategic plan to promote precision nutrition [3].

The rs671 polymorphism of the aldehyde dehydrogenase 2 (*ALDH2*) gene, an East Asian-specific polymorphism, is well known to change sensitivity to alcohol [4,5]. Carriers with minor alleles (A alleles) of the rs671 polymorphism cannot drink much alcohol because this polymorphism is translated into a non-functional aldehyde dehydrogenase [4]. Several GWASs have reported that the 12q24 region located in the rs671 polymorphism is associated not only with alcohol intake [5] but also with some dietary behaviors (DBs) such as the consumption of fish, coffee, tea, confections, and natto [6,7,8,9,10,11,12,13]. Furthermore, this polymorphism is known to be a risk factor for several cancers [14,15,16,17,18], cardiovascular disease [19,20,21], obesity [22], hypertension [23,24,25,26], and health indicators such as blood pressure [26] and BMI [27]. Therefore, the rs671 polymorphism is a critical SNP associated with DBs and disease risk in East Asians.

A three-way association among the rs671 polymorphism, DBs, and disease has been reported. Carriers of the A allele of the rs671 polymorphism with drinking habit were at a higher risk of several cancers than carriers of the G allele [15,16,17,18]. Furthermore, it has been reported that interaction between the rs671 polymorphism and consumption of fried foods in hypertension [24,25] and in coronary artery disease [21], and interactions between the polymorphism and the consumption of fresh fruits, vegetables, and salted and smoked foods in nonalcoholic fatty liver [28]. However, it is unclear whether the rs671 polymorphism genetically modifies the association between DBs and an outcome, and no information has been presented to indicate what DBs are recommended for health for each rs671 genotype.

This study aimed to determine whether rs671 genetically modifies the association between DBs and health and to provide information for precision nutrition leading to better health. Thus, in this study, 12,271 Japanese participants were tested for genetic heterogeneity of the rs671 polymorphism and the association with DBs and BMI. We further investigated the genotype and drinking habit-specific association between DBs and BMI to provide information for precision nutrition.

## 2. Materials and Methods

### 2.1. Study Participants

Participants in this study were users of “HealthData Lab”, which is a direct-to-consumer genetic testing service provided in Japan by Yahoo! Japan Corporation (Tokyo, Japan) and Genequest Inc. (Tokyo, Japan). The participants were 18 years of age or older and were asked to complete an Internet-based questionnaire that included socio-demographic factors, lifestyle habits, and medical history at the time of registration. All participants provided written informed consent for the general use of their genotype and questionnaire data. After informing the participants of the purpose of this study, we provided the opportunity to opt out. This study was conducted under the principles expressed in the Declaration of Helsinki after gaining approval from the Genequest ethics committee.

### 2.2. Genotyping of the rs671 Polymorphism in Participants

We previously reported that the 12q24 locus, which is located in the rs671 polymorphism, is associated with several DBs [6,7,8,9]. Accordingly, to obtain the genotyping data of the rs671 polymorphism in participants, we used two platforms, the HumanCore-12 + Custom BeadChip (Illumina Inc., San Diego, CA, USA) and the HumanCore-24 + Custom BeadChip (Illumina). The DNA samples of participants were extracted from saliva using the Oragene DNA collection kit (DNA Genotek Inc., Ottawa, Ontario, Canada). Both platforms included the rs671 polymorphism.

### 2.3. QC of Genotyping Data

We used 296,675 markers included in both genotyping platforms in this QC protocol. In addition, we excluded participants with inconsistent sex data between genotypes and the questionnaire, closely related pairs among the participants (identity-by-descent method, PI_HAT > 0.1875), and those with estimated non-Japanese ancestry using PLINK (version 190b3.42) [29,30] and Eigensoft (version 6.1.3) software [31]. To calculate principal components for covariates, we excluded SNPs with low call rates (<0.95), low Hardy–Weinberg equilibrium exact test *p*-values (<1 × 10^−6^), or low minor allele frequencies (<0.01) and performed principal component analysis using Eigensoft.

### 2.4. Phenotypes Based on Data Analyses

Participant age, sex, BMI, and a total of 56 DB items (including two alcohol intake parameters, 11 non-alcoholic beverage items, and 43 food intake metrics) were obtained from a web-based questionnaire. In addition, participants responded to 5–8 items based on the frequency of their monthly food or beverage intake. Categorical terms of the DBs, intake frequency, and scoring are shown in Appendix A. Alcohol consumption (amount of alcohol in grams per day) was calculated as the total amount of beer, red wine, white wine, highballs/cocktails, rice wine, and distilled spirits.

### 2.5. Statistical Analyses

We performed linear regression analysis of the association between 56 DBs and BMI with an adjustment for age and population stratification (five principal components) using PLINK 1.9. Additionally, we carried out the linear regression analysis of the association between the rs671 polymorphism and 56 DBs with an adjustment for age and population stratification with or without alcohol intake frequency and alcohol consumption as covariates. Furthermore, we performed linear regression analysis of the associations between 56 DBs and BMI stratified by the rs671 genotypes and drinking habits using R v3.2.1 [32]. We defined participants whose alcohol intake frequency was more than one time per month as drinkers, whereas the others were treated as non-drinkers. To test for genetic heterogeneity, interactions with the rs671 genotype were assessed using METAL (version 2011-03-259) [33]. Because of the limited number of participants carrying the AA genotype of the rs671 polymorphism among drinkers (31 males and 10 females), we filtered out these participants in the drinking-stratified analyses. Finally, we performed a backward-forward stepwise analysis between 56 DBs and BMI stratified by rs671 genotypes and drinking habits using R.

## 3. Results

### 3.1. Characteristics of Participants

Characteristics of our participants are shown in Table 1. In this study, 12,271 Japanese individuals, including 6525 males and 5746 females, were enrolled. The mean age ± SD was 50.61 ± 13.49 for males and 49.86 ± 12.73 for females. BMI was average for middle-aged Japanese individuals (23.89 ± 3.44 for males, 22.14 ± 3.79 for females) compared to the National Health and Nutrition Survey in 2015. The ratios of rs671 genotypes were comparable to those for all Japanese individuals as follows: 53.2% (GG), 39.2% (GA), and 7.6% (AA) in males; 55.7% (GG), 37.7% (GA), and 6.6% (AA) in females. Both alcohol frequency (times/week, GG: 3.67 ± 2.79, GA: 2.05 ± 2.57, AA: 0.08 ± 0.38 in males; GG: 2.27 ± 2.58, GA: 0.88 ± 1.82, AA: 0.03 ± 0.11 in females) and consumption (g/week, GG: 13.57 ± 15.37, GA: 6.09 ± 10.48, AA: 0.11 ± 0.60 in males; GG: 6.32 ± 10.06, GA: 2.07 ± 6.17, AA: 0.02 ± 0.13 in females) were higher in participants with the G-allele than in those with the A-allele for both sexes (*p* trend < 0.001; Table 1). Furthermore, alcohol frequency and consumption were higher in males than in females (*p* interaction < 0.001; Table 1). The means of 54 DB scores (± SD), including 11 non-alcohol beverage intake (cups/week) and 43 food intake (times/week) scores, based on the rs671 genotype, are shown in Appendix A.

### 3.2. Associations between DBs and BMI

To clarify whether each DB was associated with BMI, we performed a liner regression analysis and found that 18 and 21 DBs were significantly (*p* < 8.9 × 10^−4^, after Bonferroni correction) associated with BMI in males and females, respectively (Figure 1). Next, a liner regression analysis revealed that the rs671 polymorphism was significantly (*p* < 8.9 × 10^−4^) associated with 11 and seven DBs in males and females, respectively (Appendix A). In addition, four DBs in males and two DBs in females were still significantly (*p* < 8.9 × 10^−4^) associated with the rs671 polymorphism (Appendix A), even after adjusting this association for alcohol consumption and alcohol frequency. These results led us to speculate that the association between DBs and BMI is mediated by the rs671 polymorphism and that the rs671 modification might be both dependent on and independent of drinking habits.

### 3.3. Modification of rs671 Polymorphism with Respect to the Association between DB and BMI

To examine modification of the rs671 polymorphism, linear regression analysis and heterogeneity tests, using interactions based on the rs671 genotype, were performed with and without stratification by drinking habits. There were genetic heterogeneities (*p* < 0.05) across all analysis groups (Figure 2 and Figure 3), suggesting that the rs671 polymorphism has a genetic influence on DBs independent of drinking behavior.

### 3.4. rs671 Genotype-Specific Associations between DBs and BMI

We then carried out linear regression analysis using the stepwise variable selection method to determine the association between DBs and BMI for each drinking habit and the rs671 genotype, because this could lead to precision nutrition for weight management (Figure 4 and Figure 5). In drinkers, alcohol consumption was associated with BMI only in males and not in females. In contrast, alcohol frequency was negatively associated with BMI in both sexes. In non-drinkers, compared with that in drinkers, the association between DBs and BMI was not similar between rs671 genotypes. In females, there was a characteristic negative association between the intake frequencies of soy products (natto & soy, tofu, soy milk) and BMI. Of interest, *Lactobacillus* beverages were positively associated with BMI in participants with the AA genotype, among the rs671 polymorphisms, for both sexes. In most groups, fried foods, processed meat, soft drinks, Chinese tea, and coffee were positively associated with BMI, whereas vegetables were negatively associated.

## 4. Discussion

The results of this study show that the rs671 polymorphism genetically modifies the association between DBs and BMI. Recently, interactions between the rs671 polymorphism and DBs have been reported in several diseases, such as hypertension [23,24,25], coronary artery disease [21], and non-alcoholic fatty liver disease [28]. However, the three-way association among the rs671 polymorphism, DBs, and BMI as a health indicator in general population remained unknown. Thus, our finding suggests that the rs671 polymorphism may be a genetic modifier of DBs and health at a stage before disease onset. We further indicated the genotype and drinking habit-specific DBs associated with BMI, which can be used for the application of precision nutrition in East Asians.

Among drinkers, there were few participants with the AA type of the rs671 polymorphism because they cannot degrade aldehydes. In drinkers with GA and GG types, alcohol consumption was positively associated with BMI, whereas drinking frequency was negatively associated with BMI. This contradictory effect of alcohol consumption and frequency on BMI has been reported in previous studies [34,35,36]. It has been inferred that alcohol frequency is negatively correlated with BMI because those with moderate drinking habits have healthier lifestyles and are less likely to be overweight [36]. By contrast, large amounts of alcohol intake at one time enhance susceptibility to increases in BMI owing to the caloric intake of carbohydrates from alcohol [35]. Therefore, drinkers need to know the right amount of alcohol for their genotype and drink in a healthy way.

In non-drinkers, the characteristics of the association between DBs and BMI differed among genotypes of the rs671 polymorphism. Notably, for the AA type in both sexes, positive associations were found between the frequency of drinking *Lactobacillus* beverages and BMI. Moreover, although not significant, there was a positive association between the frequency of *Lactobacillus* beverage intake and BMI in male non-drinkers of the GA type and in female drinkers of the GG type. *Lactobacillus* beverages contain *Lactobacillus* species known for their anti-obesity effects [37,38,39]. Yang et al. reported that in *Aldh2*-knock-in mice fed a high-fat diet, those with the GA type showed significantly increased body weight and white fat mass and decreased levels of stool lactobacilli compared to those in mice with the GG type (the AA type has not been examined) [40]. From this report, it could be inferred that administering *Lactobacillus* to the AA type of mice would decrease body weight and white fat mass. However, the present results differed from this expectation. Possible reasons for this include differences between humans and mice and differences in the food environment, such as the induction of obesity via the administration of a high-fat diet in mice. Alternatively, *Lactobacillus* products might only be a proxy for other diets or participants with a higher BMI might have consumed the *Lactobacillus* beverage for its anti-obesity effects. Further studies, including more detailed diet surveys and gut microbiota analyses based on large populations, are needed to elucidate these causes.

It is worth mentioning that there was a female-specific negative association between the intake frequencies of soy products (natto & soy, tofu, soy milk) and BMI. Genistein, which is one of the isoflavones that is abundant in soybeans, has a reported estrogen-like effect [41,42,43]. Because estrogen decreases visceral fat accumulation via estrogen receptors [44], menopausal women have increased visceral fat [45]. As the participants in this study were close to menopausal age, it is possible that their estrogen levels were lower than those in a young population and that the anti-obesity effects of soy product consumption were accordingly more obvious.

There are several limitations to this study. First, this study was a questionnaire-based food intake frequency (FFQ) survey. Thus, energy adjustment and nutrient estimation were not possible. In addition, we could not determine what was consumed with the food being studied, such as sugar and milk in coffee. Furthermore, our previous results of GWASs to examine the association between genetic loci and DBs [6,7,8] was in agreement with GWASs on other Japanese groups [10,11,12]. Therefore, we consider that our FFQ can at least be used to determine the association with DBs. In the future, the results of this study should be replicated with other populations. Second, this study was conducted only on Japanese individuals. Therefore, it is not clear whether the results of this study could be extrapolated to East Asians with the rs671 polymorphism other than Japanese. Accordingly, replication will be necessary to examine this association in detail. Finally, since the rs671 polymorphism is located in the 12q24 region, a linkage disequilibrium locus, we cannot exclude the possibility that other SNPs might modify the phenotypic association or effects of rs671. Thus, studies using human rs671-transgenic mice or cell lines are needed to determine the direct effect of the rs671 polymorphism in the future.

## 5. Conclusions

In conclusion, we show that the rs671 polymorphism modifies associations between DBs and BMI. Moreover, these modifications have characteristics that depend on sex and drinking habits. These findings provide evidence to support the possibility of using rs671 polymorphism information for precision nutrition for East Asians.

## Figures and Tables

**Figure 1 nutrients-14-05116-f001:**
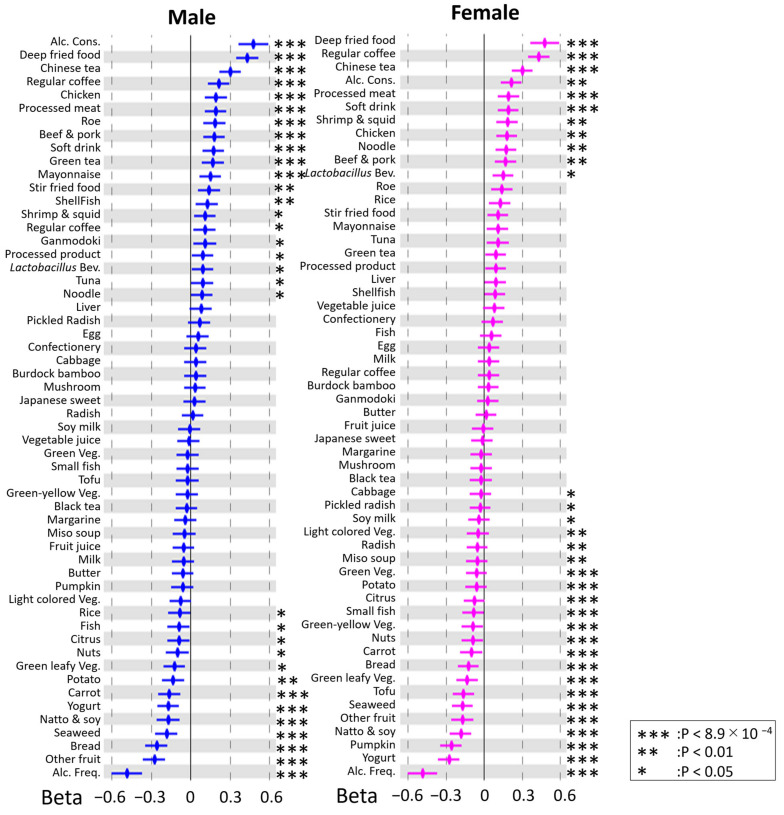
Forest plot showing association between dietary behaviors (DBs) and BMI. The *X*-axis and *Y*-axis show the standardized beta coefficient  ±  standard deviation and each DB, respectively. Blue and pink spots indicate the results for males and females, respectively. Alc. Cons, alcohol consumption; Alc. Freq., alcohol frequency; *Lactobacillus* Bev., *Lactobacillus* beverage; Green Veg, green vegetable; Green leafy Veg., green leafy vegetable; Green-yellow Veg, green-yellow vegetable; Light-colored Veg., light-colored vegetable. Asterisks indicate statistical significance (***, *p*  <  8.9 × 10^−4^; **, *p*  < 0.01; *, *p* < 0.05).

**Figure 2 nutrients-14-05116-f002:**
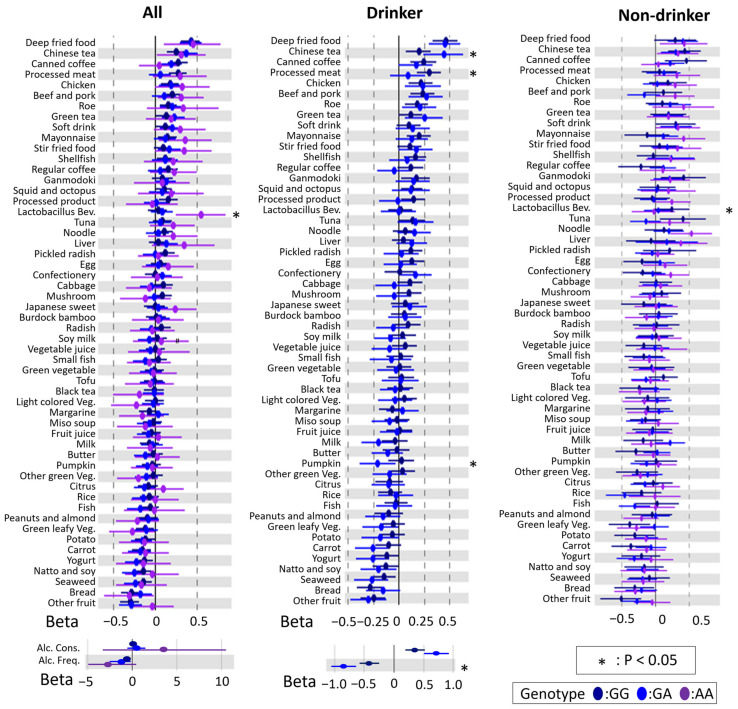
Forest plot showing association between dietary behaviors (DBs) and BMI modified by rs671 genotype in males. The *X*-axis and *Y*-axis show the standardized beta coefficient  ±  standard deviation and each DB, respectively. Dark blue, blue, and purple spots indicate the results for GG, GA, and AA types of rs671 genotypes, respectively. Alc. Cons, alcohol consumption; Alc. Freq., alcohol frequency; *Lactobacillus* Bev., *Lactobacillus* beverage; Green Veg, green vegetable; Green leafy Veg., green leafy vegetable; Green-yellow Veg, green-yellow vegetable; Light-colored Veg., light-colored vegetable. Asterisks indicate statistical significance for heterogeneity (*, *p* < 0.05).

**Figure 3 nutrients-14-05116-f003:**
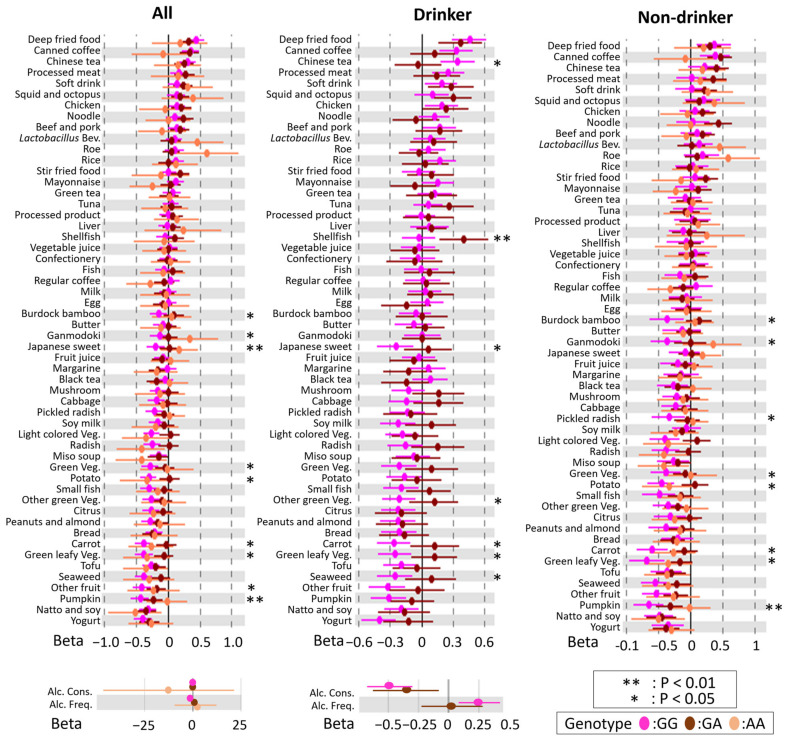
Forest plot showing association between dietary behaviors (DBs) and BMI modified by rs671 genotype in females. The *X*-axis and *Y*-axis show the standardized beta coefficient  ±  standard deviation and each DB, respectively. Pink, blown, and beige spots indicate the results for GG, GA, and AA types of rs671 genotypes, respectively. Alc. Cons, alcohol consumption; Alc. Freq., alcohol frequency; *Lactobacillus* Bev., *Lactobacillus* beverage; Green Veg, green vegetable; Green leafy Veg., green leafy vegetable; Green-yellow Veg, green-yellow vegetable; Light-colored Veg., light-colored vegetable. Asterisks indicate statistical significance for heterogeneity (**, *p*  < 0.01; *, *p* < 0.05).

**Figure 4 nutrients-14-05116-f004:**
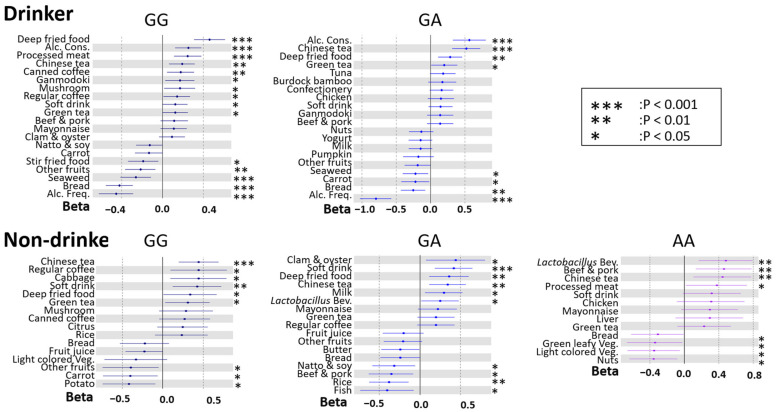
Forest plot showing associations between dietary behaviors (DBs) and BMI for each rs671 genotype in males. The *X*-axis and *Y*-axis show the standardized beta coefficient  ±  standard deviation and each DB, respectively. Dark blue, blue, and purple spots indicate the results for GG, GA, and AA types of rs671 genotypes, respectively. Alc. Cons, alcohol consumption; Alc. Freq., alcohol frequency; *Lactobacillus* Bev., *Lactobacillus* beverage; Green leafy Veg., green leafy vegetable; Light-colored Veg., light-colored vegetable; Green-yellow Veg, green-yellow vegetable. Asterisks indicate statistical significance for heterogeneity (***, *p*  <  0.001; **, *p*  < 0.01; *, *p* < 0.05).

**Figure 5 nutrients-14-05116-f005:**
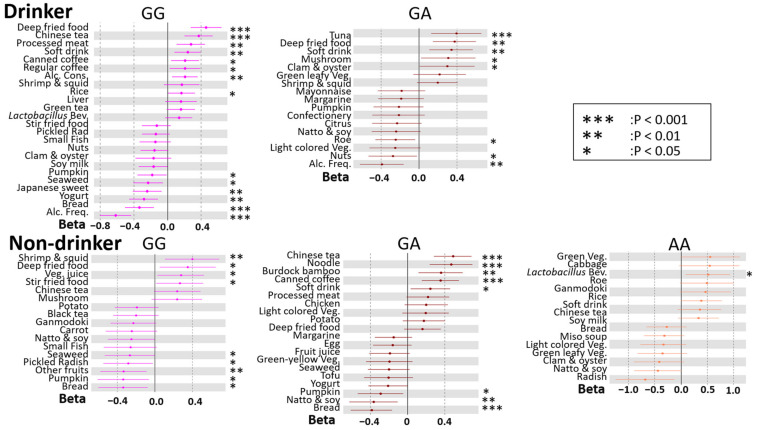
Forest plot showing association between dietary behaviors (DBs) and BMI for each rs671 genotype in females. The *X*-axis and *Y*-axis show the standardized beta coefficient  ±  standard deviation and each DB, respectively. Pink, blown, and beige spots indicate the results for GG, GA, and AA types of rs671 genotypes, respectively. Alc. Cons, alcohol consumption; Alc. Freq., alcohol frequency; *Lactobacillus* Bev., *Lactobacillus* beverage; Light colored Veg., light-colored vegetable. Asterisks indicate statistical significance for heterogeneity (***, *p*  <  0.001; **, *p*  < 0.01; *, *p* < 0.05).

**Table 1 nutrients-14-05116-t001:** Characteristics of participants.

	Male						Female						*p* _interaction_ ^c^
	Genotype ^a^			Beta	SE	*p* Trend ^b^	Genotype ^a^			Beta	SE	*p* Trend ^b^	
	GG	GA	AA				GG	GA	AA				
Number	3474	2556	495				3203	2163	380				
Age, years ^d^	50.41 ± 13.58	50.71 ± 13.4	51.49 ± 13.31	0.43	0.26	0.10	49.46 ± 12.76	50.35 ± 12.61	50.49 ± 13.01	0.7	0.27	0.01	0.48
BMI, kg/m^2 d^	23.96 ± 3.38	23.88 ± 3.53	23.53 ± 3.33	−0.16	0.07	0.002	22.25 ± 3.85	21.98 ± 3.70	22.05 ± 3.67	−0.18	0.08	0.02	0.82
Alcohol intake ^d^													
Alcohol frequency, times/week	3.67 ± 2.79	2.05 ± 2.57	0.08 ± 0.38	−1.71	0.05	<0.001	2.27 ± 2.58	0.88 ± 1.82	0.03 ± 0.11	−1.25	0.05	<0.001	<0.001
Alcohol frequency, times/week	13.57 ± 15.37	6.09 ± 10.48	0.11 ± 0.60	−7.06	0.25	<0.001	6.32 ± 10.06	2.07 ± 6.17	0.02 ± 0.13	−3.68	0.18	<0.001	<0.001

^a^ Genotype of the rs671 polymorphism in *ALDH2* gene. ^b^
*p*-value of linear regression analysis with rs671 genotype. ^c^
*p*-value of interaction between sex and rs671 genotype using linear regression analysis. ^d^ Mean ± SD values of measured participants.

## Data Availability

The data generated or analyzed in this study are included in this published article and its Appendix A. Other data are available from the authors on reasonable request.

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
