# Peer review of "Association between Dietary Behaviors and BMI Stratified by Sex and the ALDH2 rs671 Polymorphism in Japanese Adults"

_nutrients, 2022, doi:10.3390/nu14235116_

Round 1
Reviewer 1 Report
Thank you for the opportunity to review your manuscript. It presents an extremely pertinent and important topic. Nevertheless, I have a few reservations.
- The introduction is too short, currently presenting neither the current knowledge nor the research gap that the authors wanted to address.
- the objective lacks a hypothesis or research question.
- the tables do not include results for statistical tests and for correlation coefficients.
Author Response
We appreciate these comments. Following them, we modified our manuscript. The modified parts of the text are shown in red.
Comment: Thank you for the opportunity to review your manuscript. It presents an extremely pertinent and important topic. Nevertheless, I have a few reservations.
- The introduction is too short, currently presenting neither the current knowledge nor the research gap that the authors wanted to address.
Response: Thank you for this important comment. Following this comment, we modified and added a paragraph in the introduction (lines 43-56). In addition, we modified in Discussion (lines 186-192).
Along with those, we have added 11 citations.
Comment: - the objective lacks a hypothesis or research question.
Response: We are grateful for this comment. We modified the introduction section (lines 57-60).
Comment: - the tables do not include results for statistical tests and for correlation coefficients.
Response: Thank you for this comment. As per the comment, we added beta and SD in Table 1 and statistical methods in the footnote.

Reviewer 2 Report
I read your article with interest. Your desription is exellent and it is well written and easy to read. You adress the shortcommings of your study. Nonetheles I feel these findings are significant and warrant publication.
Author Response
We are grateful for the favorable comments. We are very honored that you considered our research worth publishing, even in light of some limitations of our research.